# The Therapeutic Effect of Phosphopeptide P140 Attenuates Inflammation Induced by Uric Acid Crystals in Gout Arthritis Mouse Model

**DOI:** 10.3390/cells11233709

**Published:** 2022-11-22

**Authors:** Izabela Galvão, Dylan Mastrippolito, Laura Talamini, Mariana Aganetti, Victor Rocha, Cindy Verdot, Viviani Mendes, Vivian Louise Soares de Oliveira, Amanda Dias Braga, Vinicius Dantas Martins, Ana Maria Caetano de Faria, Flávio A. Amaral, Philippe Georgel, Angélica T. Vieira, Sylviane Muller

**Affiliations:** 1Laboratory of Microbiota and Immunomodulation, Institute of Biological Sciences, Department of Biochemistry and Immunology, Universidade Federal de Minas Gerais (UFMG), Belo Horizonte 31270-901, Brazil; 2CNRS UMR7242, Biotechnology and Cell Signaling/Strasbourg Drug Discovery and Development Institute (IMS), University of Strasbourg, 67000 Strasbourg, France; 3Experimental Rheumatology Laboratory, Immunopharmacology Group, Department of Biochemistry and Immunology, and Institute of Biological Sciences, Federal University of Minas Gerais, Belo Horizonte 31270-901, Brazil; 4Laboratory of Immunobiology, Department of Biochemistry and Immunology, Institute of Biological Sciences, Universidade Federal de Minas Gerais (UFMG), Belo Horizonte 31270-901, Brazil; 5ImmunoRhumatologie Moléculaire, UMR_S 1109, INSERM, University of Strasbourg, 67000 Strasbourg, France; 6Fédération de Médecine Translationelle de Strasbourg (FMTS), 67000 Strasbourg, France; 7Institute for Advanced Study, University of Strasbourg, 67000 Strasbourg, France

**Keywords:** MSU crystal-induced inflammation, pain, inflammasome, neutrophil, autophagy, peptide therapy

## Abstract

Gout is a painful form of inflammatory arthritis characterized by the deposition of monosodium urate (MSU) crystals in the joints. The aim of this study was to investigate the effect of peptide P140 on the inflammatory responses in crystal-induced mouse models of gout and cell models including MSU-treated human cells. Injection of MSU crystals into the knee joint of mice induced neutrophil influx and inflammatory hypernociception. Injection of MSU crystals subcutaneously into the hind paw induced edema and increased pro-inflammatory cytokines levels. Treatment with P140 effectively reduced hypernociception, the neutrophil influx, and pro-inflammatory cytokine levels in these experimental models. Furthermore, P140 modulated neutrophils chemotaxis in vitro and increased apoptosis pathways through augmented caspase 3 activity and reduced NFκB phosphorylation. Moreover, P140 increased the production of the pro-resolving mediator annexin A1 and decreased the expression of the autophagy-related ATG5-ATG12 complex and HSPA8 chaperone protein. Overall, these findings suggest that P140 exerts a significant beneficial effect in a neutrophilic inflammation observed in the model of gout that can be of special interest in the design of new therapeutic strategies.

## 1. Introduction

Gouty arthritis is a disease of the joints that affects 1–2% of the adult population worldwide. It is a common crystal-induced arthritis, in which monosodium urate (MSU) crystals precipitate within joints and soft tissues and elicit an inflammatory response characterized by swelling, redness, and severe attacks of pain [1]. 

Acute gouty inflammation is initiated by recognition of MSU crystals by resident cells that produce pro-inflammatory mediators, especially IL-1β, which is released by activation of the nucleotide-binding domain leucine-rich-containing family pyrin domain-containing-3 (NLRP3) inflammasome [2,3,4]. Subsequent to the inflammasome activation, there is increased infiltration of neutrophils and production of pro-inflammatory mediators such as cytokines in the joints. Neutrophils are the main inflammatory cells recruited to the joint in a CXC motif chemokine receptor 1/2-dependent manner; they significantly contribute to amplifying the tissue damage [4,5]. The NLRP3 inflammasome, a key tissue damage sensor, is a dominant driver of various autoinflammatory and autoimmune diseases, including gout, rheumatoid arthritis, and lupus [6]. Of note, studies suggest that autophagy, a cellular waste removal and rejuvenation process, plays a vital role as a macrophage-intrinsic negative regulator of NLRP3 inflammasome [7]. The role of autophagy and its dysregulation in gout and the possible impact of soluble or precipitated uric acid in modulating autophagy flux remains poorly understood [8,9]. 

Although gout has been known since ancient times, the prevalence of gout has risen sharply within the last decades, rendering this disease a critical healthcare issue [10]. The current therapies are not well tolerated in all patients, especially those with longstanding hyperuricemia, comorbidities, and contraindications [1]. Novel therapeutic approaches are therefore eagerly awaited to improve the existing strategies or to provide additional possibilities to treat gout patients. In this context, we propose to evaluate the effect of synthetic phosphopeptide P140, which was found to display potent effects in several murine models of autoimmunity, including a murine model of secondary Sjögren’s syndrome, experimental autoimmune neuritis, and experimental colitis [11,12,13] and patients with systemic lupus erythematosus [14,15,16,17]. P140 is a 21 amino acid residue-long peptide derived from the 131–151 sequence of the snRNP U1-70K spliceosomal protein [16,18]. Most of the effects of P140 were attributed to its ability to regulate dysfunctional autophagy processes, particularly in the lupus model using MRL/lpr mice [17,19,20]. Its effects essentially target lysosomes and lysosomal-dependent autophagy but not mitochondria and mitophagy [10]. Recently, the anti-inflammatory effect of P140 has been demonstrated further in a murine model of asthma, in which it prevented lung dysfunctions and the recruitment of inflammatory cells, especially eosinophils and neutrophils, into the bronco-alveolar fluids [21]. In this study, we have evaluated the effects of P140 in two different mouse models of gout. 

## 2. Materials and Methods

### 2.1. Animals and Ethics Statement

Experiment protocols involving animals were approved by the respective local institutional animal care and use committees in Brazil by the Universidade Federal de Minas Gerais (# 309/2019); in France, by the Comité Régional d’Éthique pour l’Expérimentation Animale de l’Université de Strasbourg and the French ministry for higher education, research, and innovation (APAFiS # 201808301413304127). In line with these agreements and considering the best practices in the field (3R’s rule), we took the necessary measures to avoid pain and minimize distress and pointless suffering of mice during experiments and at the time of sacrifice. Animals were supplied with water and food ad libitum. They were maintained under controlled environmental conditions (20 ± 2 °C) in either specific pathogen-free or conventional husbandry conditions. A 12 h/12 h light–dark cycle (lighting 7:00 a.m.–7:00 p.m.) was maintained. Mice were housed in large polycarbonate cages, with 8 to 10 mice per cage on bedding made from spruce wood chips and enriched with play tunnels changed weekly. 

### 2.2. Mice and MSU Crystal-Induced Gout 

MSU crystals used in Brazil were prepared by dissolving uric acid (5 mg/mL) in 0.1 M borate buffer (pH 8.5) as previously described [3]. MSU crystals used in France were generated as described [22]. The MSU crystals used were endotoxin-free as determined by the Limulus amoebocyte lysate assay. Male C57Bl/6 mice (8–12 weeks) were obtained from the Centro de Bioterismo of the Universidade Federal de Minas Gerais in Brazil and Janvier labs in France. Two mouse models were used in this study. In the first model, mice received an intra-articular (i.a.) injection of either 100 μg of MSU crystals diluted in 10 μL of phosphate-buffered saline (PBS; pH 7.2) or PBS alone (namely control) [4]. Mice were euthanized 15 h after the MSU challenge. The knee cavity was washed with PBS (2 × 5 µL) to collect the cells. Periarticular tissues were collected to evaluate the levels of cytokines and chemokines, and myeloperoxidase (MPO) activity. Blood was collected through the brachial plexus and total cells were counted using a Neubauer chamber and smears were prepared for leukocyte counts. The total number of leukocytes from knee wash was determined similarly by counting cells in a Neubauer chamber after staining with Turk’s solution. Differential cell counts were determined using standard morphological criteria and were performed on May–Grunwald–Giemsa-stained slides. In the second model of gout, a single injection of 3 mg MSU crystals was given to mice subcutaneously (s.c.) into the hind paw [23]. Control mice were injected s.c. with PBS alone. At 24 h, the thickness of the hind paw was measured using a calliper. At the end of the experiments, all mice were sacrificed under gas anaesthesia (isoflurane). The hind paws were collected for further study.

### 2.3. Experimental Design 

In one set of experiments, mice received an injection intravenously of either the P140 peptide or the scrambled ScP140 used as the peptide control at a concentration of 100 μg/mouse/injection and 1 h after treatment mice were placed under anaesthesia (80:15 mg/kg ketamine:xylazine, i.p. Syntec, Sao Paulo, Brazil) and were injected with MSU crystals into the tibiofemoral joint. After 15 h of MSU injection, mice were euthanized, and inflammatory parameters were evaluated. 

In the second set of experiments a single injection of MSU crystals (s.c. 3 mg) into the hind paw was performed. Mice received at the same time of MSU injection an injection intravenously of either the P140 peptide or the ScP140 used as the peptide control at a concentration of 100 μg/mouse/injection. After 24 h of MSU injection, mice were euthanized, and inflammatory parameters were evaluated. 

In the last set of experiments, animals were injected with MSU crystals into the tibiofemoral joint and after 6 h of MSU injection mice were post-treated with P140 peptide (100 μg/mouse i.v.) or dexamethasone (2 mg/kg; intraperitoneally (i.p.). Sigma-Aldrich D4902, St. Louis, MO, USA) used as positive control. After 15 h of MSU injection, mice were euthanized, and inflammatory parameters were evaluated. The timelines of the treatments of each experiment are displayed in the respective figure. 

Number of animals used in this study were calculated using a statistical software (GPOWER 3.1.9.2), considering the variation of number of neutrophils in the knee lavage in our previous publications. We detailed the number of animals in the figure legends of each graph. Animals were not randomized since they are all genetically identical.

### 2.4. Synthetic Peptides

P140 (RIHMVYSKRpSGKPRGYAFIEY) and Sc140 (YVSRYFGpSAIRHEPKMKIYR) phosphopeptides (where pS represents phosphoserine residues) were synthesized as described previously [24] using classical N-[9-fluorenyl] methoxycarbonyl solid-phase chemistry and purified by reversed-phase high-performance liquid chromatography (RP-HPLC). Peptide homogeneity was checked by analytical HPLC, and their identity was assessed by liquid chromatography-mass spectrometry on a Finnigan LCQ Advantage Max system (Thermo Fischer Scientific, Waltham, MA, USA). The lyophilized peptides were first solubilized in distilled water and then to the desired concentration in culture medium or other buffers for in vitro experiments. For animal administration, they were diluted in 0.9% (*w/v*) NaCl. 

### 2.5. Evaluation of Hypernociception

Mechanical hypernociception was tested in mice as reported previously [25]. Briefly, mice were placed in acrylic cages with a wire grid floor and an electronic pressure meter was used (anesthesiometer; electronic von Frey, Insight Instruments, Ribeirão Preto, SP, Brazil). The electronic pressure-meter apparatus automatically recorded the intensity of the force applied when the paw was withdrawn. The less pressure applied to the paw the more inferred behavioural pain response is observed. The term “hypernociception” was used to define the decrease of nociceptive withdrawal threshold in experimental animals [26]. Results are expressed as withdrawal threshold (in g) calculated by the interval mean measurements from the bilateral joint.

### 2.6. Cytokines and Myeloperoxidase Determination

ELISA kits from R&D Systems were used to quantify the concentrations of CXCL1 (Cat.No. MKC00B) and IL-1β (Cat.No. MLB00C) in the supernatant of knee tissue; and MPO (Cat.No. DY3667), IL-1β (Cat.No. MLB00C), and TNF-α (Cat.No. MTA00B) in hind paws. The knee tissue was homogenized as follows: for each 100 mg of tissue, 1 mL of PBS (0.4 M NaCl and 10 mM NaPO_4_) containing protease inhibitors (0.1 mM phenylmethylsulphonyl fluoride, 0.1 mM benzethonium chloride, 10 mM ethylenediaminetetraacetate/EDTA and 20 mM aprotinin A/pancreatic trypsin inhibitor) and 0.05% (*v/v*) Tween-20 was added. The samples were then centrifuged for 10 min at 3000× *g,* and the supernatants immediately used for ELISA tests. The results were expressed as pg per mg of tissue. For MPO enzymatic assay, the pellets from samples homogenized for cytokine measurements and spleens were homogenized with 0.05 M NaH_2_PO_4_ containing 0.5% (*v/v*) of hexadecyltrimethylammonium bromide (Sigma-Aldrich, Cat.No. H5882). Samples were frozen 3 times in liquid nitrogen and centrifuged to collect the supernatant for MPO assay. The final reaction was visualized by adding H_2_O_2_ as peroxidase substrate and 3,3′,5,5′-tetramethylbenzidine as chromogen, for 30 min at 37 °C. The reaction was blocked by adding 1M HCl, before measuring absorbance at 450 nm using a plate reader (Thermo Scientific Multiskan GO Microplate Spectrophotometer). Results were expressed as absorbance values.

### 2.7. Human Neutrophils Isolation and Purification 

Neutrophils were obtained from heparinized peripheral blood of healthy human donors (Etablissement Français du Sang, Strasbourg, France). Neutrophils were isolated by negative selection using the MACSxpress kit for the isolation of human neutrophils from whole blood (Miltenyi Biotec, Cat.No. 130-104-434, Paris, France) and the magnetic separator (Miltenyi Biotec, Cat.No. 130-098-308, Paris, France). The supernatants containing neutrophils were collected and the removal of contaminating erythrocytes was performed by positive selection using the MACSxpress human erythrocyte depletion kit (Miltenyi Biotec, Cat.No. 130-098-196, Paris, France) using the manufacturer’s instructions. Neutrophils were suspended in RMPI 1640 culture medium (Gibco, Cat.No. 11875085, Pasley, Scotland). Neutrophil viability was assessed using acridine orange/propidium iodide labelling (Logos Biosystem, Cat.No. F23001, Villeneuve-d’Ascq, France), and their counts measured using a LUNA-FL™ dual fluorescence cell counter (Logos Biosystem, Cat.No. L20001-LG, Villeneuve-d’Ascq, France). The viability of neutrophils was above 98%. After cell counting, neutrophils were incubated at a rate of 1 × 10^6^ cells/well (in 24-well plate) or 1 × 10^5^ cells/well (in 96-well plate). The culture plates were then incubated at 37 °C under 5% CO_2_.

### 2.8. Neutrophils Chemotaxis Assay with Boyden Chamber

Human neutrophils were isolated from whole blood and cultured at a density of 1 × 10^5^ cells/well (96-well plate) in RPMI 1640. Then, they were incubated for 1 h at 37 °C with 20 μM P140 peptide and stimulated first with *E. coli* LPS (500 ng/mL) and then for 4 h further with MSU (1 mg/mL). The culture media containing neutrophils were then loaded into the upper Boyden chambers (96-well plates; 3-μm pore size; Abcam cell migration kit, ab235692). Neutrophil migration was induced by adding chemokine CXCL8 (50 ng/mL in RPMI medium; R&D system, Cat.No. 208-IL) placed in the lower chambers. Then, the upper chambers were combined with the lower chambers and incubated at 37 °C for 45 min under 5% CO_2_. The plates were centrifuged at 1000× *g* for 5 min, the culture medium was removed, the cells were washed with the provided wash buffer and the plates were centrifuged again at 1000× *g* for 5 min to remove the wash buffer. Neutrophils that had migrated from the upper chamber to the lower chamber were labelled using the fluorescent staining reagent provided in the Abcam kit. The plate with the lower chambers was then incubated at 37 °C under 5% CO_2_ for 1 h. Fluorescence was measured using the PerkinElmer VICTOR X5 2030 plate reader (Ex/Em = 530/590 nm).

### 2.9. Histopathology

Samples were processed as previously described [27]. Briefly, knee joints were collected, fixed in 10% *v/v* formol, and decalcified for 30 days in 14% *w/v* ethylenediaminetetraacetic acid—EDTA. Tissues were included in paraffin, sectioned (5 µm), and stained with Haematoxylin and Eosin (H&E). The sections of knee joints were examined in the optical microscope. The parameter evaluated was intensity and extension of inflammatory infiltrate and the picture was representative and original magnification × 4.

### 2.10. Bone Marrow-Derived Macrophages Differentiation

Bone marrow-derived macrophages (BMDMs) were obtained as previously described [28]. Briefly, cells were obtained from the tibia and femur of adult C57Bl/6 mice and were incubated in Roswell Park Memorial Institute medium (RPMI) 1640 (Gibco, Cat.No. 11875085, Pasley, Scotland) supplemented with 20% (*v/v*) of foetal bovine serum (Dutscher, Cat.No. S1018-500, Bernolsheim, France) and 30% (*v/v*) of L929 conditioned medium enriched in macrophage colony stimulating factor, for 7 days at 37 °C in humidified 5% CO_2_. Differentiated BMDM were detached, counted, seeded in a tissue culture plate (1.0 × 10^6^ cells/mL) overnight and finally stimulated the following day as described below. 

### 2.11. Analysis of Elements of the Inflammasome Pathway and Chemokine Release (Caspase1, IL-1β, CXCL1)

BMDM (1.0 × 10^6^ cells/mL) were treated with peptides P140 or ScP140 (20 µM) or PBS (Untreated), 2 h before Toll-like receptor (TLR)4 agonist of lipopolysaccharide (LPS; from *Escherichia coli* serotype O:111:B4; 1 µg/mL; Sigma-Aldrich, L2630, St.Louis, MO, USA) was added for 1 h at 37 °C. Peptide P140 was also used in priming (instead of LPS 1 h before MSU stimulation). Then inflammasome activation was performed by MSU crystal (300 µg/mL) or ATP (5 mM) stimulation for further 6 h or 30 min, respectively. Supernatants were removed and analysed by western blotting as previously described [29] and enzyme-linked immunosorbent assay (ELISA) kit. Blots were probed with mouse antibodies caspase-1 p20 (Adipogen, AG-20B-0042, San Diego, CA, USA) according to the manufacturer’s instructions. Levels of chemokine (CXC motif) ligand CXCL1 and interleukin 1β (IL)-1β were measured using an ELISA kit (R&D system, Cat.No. MKC00B and DY401-05 respectively, Mineapolis, MN, USA).

### 2.12. Assessment of Neutrophil Apoptosis 

The evaluation of neutrophil morphology was carried out using the first animal model described in our study. At sacrifice, the knee cavity was washed with PBS (2 × 5 µL) to collect cells. To analyse the apoptotic morphology of these cells, cytospin preparations of neutrophils were examined after May–Grünwald–Giemsa staining. The typical morphologic changes indicating apoptosis, such as cell shrinkage and chromatin condensation-yielding fragments, were recorded in at least 100 cells per slide. The data were reported as the percentage of apoptotic neutrophils.

### 2.13. Caspase-3 Assay 

Human neutrophils were isolated from whole blood and cultured at a density of 1 × 10^5^ cells/well (96-well plate) in RPMI 1640. Neutrophils were left untreated or treated with 20µM P140 and incubated for 4 h at 37 °C under 5% CO_2_. Caspase-3 activity was measured using the Caspase-Glo 3/7 kit (Promega, Cat.No. G8090, Madison, WI, USA). After 4 h of incubation, 100 μL of a mixture of provided substrate amino luciferin and buffer (O_2_, ATP, Mg^2+^) was added to each well and the plate was stored in the dark. Empty wells were used as a negative control (blank), whose luminescence value is subtracted from the experimental values. The plate was then incubated at 37 °C for 2 h in the Varioskan Lux5 luminometer (ThermoFischer Scientific). 

### 2.14. Analysis of NF-κB Activation, Annexin A1 and Autophagy Markers 

P65 nuclear factor kappa-light-chain-enhancer of activated B cells (NFκB), annexin A1, autophagy-related (ATG)5, and 12 proteins and HSPA8 chaperone protein (target of P140 peptide) were analysed in neutrophils isolated from human peripheral blood, as described previously [30]. The protein concentration of the lysates was determined by Bradford assay reagent (Bio-Rad, Cat.No. 5000006, Hercules, CA, USA). Samples (30 µg protein) were separated by electrophoresis on a denaturing 10% sodium dodecyl sulphate-polyacrylamide gel electrophoresis (SDS-PAGE) and transferred onto nitrocellulose membranes. Membranes were blocked for 1 h with 5% (*w/v*) of non-fat dry milk solution (PBS 0.1% Tween-20) and incubated overnight with an antibody to the covalent 55-kDa ATG5-ATG12 complex (0.5 μg/mL; Cell Signaling Technology, Cat.No. 12994, Beverly, MA, USA), HSPA8 (0.5 μg/mL; Abcam, ab51052), phospho-NFκB p65 (1 μg/mL; Cell Signaling Technology, Cat.No. 3031, Beverly, MA, USA) and annexin-A1 rabbit (1 μg/mL; Invitrogen, Cat.No. 713400, Carlsbad, CA, USA). After washing, membranes were incubated with appropriate horseradish peroxidase-conjugated secondary antibody (Santa Cruz, Cat.No. sc-2357, CA, 1:3000). Immunoreactive bands were visualized by using the Clarity Western ECL transfer substrate (BioRad, Cat.No. 1705061, Hercules, CA, USA). Expression levels of proteins in neutrophil lysates were normalized by densitometry relative to the total protein level, using Image Lab (Bio-Rad) software.

### 2.15. Statistical Analyses

All results are presented as mean ± SEM. Statistical analyses were performed using GraphPad Prism (version 9.0.). The normal distribution of samples was evaluated by D’Agostino–Pearson test. Data analyses were carried out by Kruskal–Wallis and/or Mann–Whitney non-parametric test. Results were considered significant at *p* < 0.05. 

## 3. Results

### 3.1. Pre-Treatment with P140 Decreases MSU-Induced Inflammation in Two Different Murine Models of Gout 

As previously described, the injection of MSU crystals in the knee joint of mice induces a significant leukocyte recruitment at 6 h predominantly consistent with neutrophil that reached maximum number at 15 h after MSU injection. Moreover, neutrophil-dependent tissue injury and articular dysfunction is observed during inflammatory response induced by MSU crystals [4]. To examine the effects of peptide P140 on these typical features, in a first series of experiments, C57BL/6 male mice (8–12-week-old) received P140 or ScP140 peptide intravenously (100 µg per mouse) 1 h before injection of MSU crystals in the knee joint of mice (Figure 1A). After 15 h of MSU injection, mice that had not received any treatment (untreated) showed redness and swelling in the knee whereas mice pre-treated with P140 showed decreased inflammatory signs as visualized in the image (Figure 1A), and joint dysfunction as assessed by measuring hypernociception (Figure 1B). Furthermore, there was significantly reduced infiltration of total inflammatory cells, mostly neutrophils, in the knee cavity of mice pre-treated with P140 when compared to untreated MSU-challenged mice (Figure 1C). In parallel, there was a reduction of inflammatory infiltrate in the histology (Appendix A), MPO activity (Mean ± SD: PBS = 0.197 ± 0.113; MSU = 0.511 ± 0.147; MSU + P140 = 0.309 ± 0.256), CXCL1 production activity (mean ± SD: PBS = 351.4 ± 58.79; MSU = 461.6 ± 45.32; MSU + P140 = 404.8 ± 99.23), and IL-1β production in periarticular tissue (Figure 1D). No statistically significant difference was observed between the untreated MSU group and the MSU group pre-treated with ScP140 that was used as a peptide control (Figure 1B–D). 

In a second set of experiments, we tested the effect of P140 given simultaneously with MSU (Figure 1E). In this model, MSU crystals were injected s.c. on the dorsal face of the hind paws of 8-week old C57BL/6 male mice [23]. The inflammation induced by MSU in the mouse’s hind paw was visible, characterized by edema and swelling as shown by paw thickness in Figure 1E, and accompanied by increased production of inflammatory markers, Il-1β and TNF-α, and MPO activity. Such effects were absent in contralateral paws (controls) (Figure 1E,F). As above, P140 significantly reduced the inflammation confirmed by paw thickness (Figure 1E), MPO activity, and levels of Il-1β and TNF-α cytokines (Figure 1F). ScP140 showed no significant effect except for MPO activity.

Next, we examined the effect of P140 on neutrophils in the blood. As shown in Figure 2, mice receiving an i.a. injection of MSU increased the number of both total cells and peripheral neutrophils compared to control mice not injected with MSU (Figure 2A,B, dashed line). However, it is noteworthy that P140 strongly raises the levels of neutrophils in the blood, as well as in the spleen (mean ± Standard Deviation (SD): PBS = 0.130 ± 0.06; MSU = 0.627 ± 0.06; P140 = 1.162 ± 0.935), of mice challenged with MSU compared to MSU-untreated mice (Figure 2B). These results combined with those shown in Figure 1C support the view that P140 might interfere with neutrophil influx to the site of inflammation. To confirm this possible effect of P140 on neutrophil migration, we conducted in vitro chemotaxis assay by using neutrophils isolated and purified from the blood donors and stimulated with MSU and LPS. In a statistically significant manner, P140 effectively decreased the CXCL8-induced migration of neutrophils in vitro (Figure 2C).

Altogether, these findings suggest that preventive therapy with P140 can exert an immunomodulatory effect on the local inflammatory response induced by MSU crystals by reducing neutrophil migration and pro-inflammatory cytokine production, particularly IL-1β. 

### 3.2. In Vitro Treatment with P140 Does Not Affect the Inflammasome Activation in BMDM Derived from Mice

The gout inflammatory response initiates during phagocytose of MSU crystals by resident macrophages with subsequent activation of the inflammasome, IL-1β, and other pro-inflammatory cytokine release [31]. These cytokines are essential to recruiting neutrophils, which participate in the progression of inflammation. As we observed a reduction in neutrophils in the site of inflammation of mice pre-treated with P140, we further investigated the effect of P140 on macrophages by using in vitro culture of mouse BMDM. As observed in Appendix A, the pre-treatment with P140 peptide 2 h before LPS priming increased caspase-1 release after MSU crystals. Of note, using P140 peptide as a priming step before MSU or ATP stimulation did not induce caspase-1 release, suggesting that P140 peptide seems to cause an increase in inflammasome activity in primed macrophages, not in resting macrophages. Cells pre-treated with P140 and then stimulated with LPS and MSU still showed an increase in CXCL1 levels and partially IL-1β (Appendix A).

Together, these results suggest that the impact of P140 on MSU-induced neutrophil recruitment might be independent of direct effects on resident macrophages but instead focuses on neutrophil activity.

### 3.3. Post-Treatment of P140 Does Not Significantly Affect Neutrophil Accumulation When an Inflammatory Status Occurs 

To evaluate the physiological effect of P140 on neutrophils during the outcome of MSU crystal-induced inflammation, we treated mice with P140 after the establishment of inflammation induced by MSU crystal injection in the knee (6 h after MSU injection). Using this schedule of treatment, we observed that P140 didn’t exhibit a significant effect on the total numbers of leukocytes (Figure 3A), neutrophils (Figure 3B), and mononuclear cells (Figure 3C), even though a trend of decrease in inflammation is observed similar to dexamethasone (Dexa)–treated mice (used as a potent anti-inflammatory and pro-resolving drug). Moreover, P140 administered post-MSU induction reduced the level of nociception measured by paw pression (mean ± SD: PBS = 8.62 ± 0.601; MSU = 2.88 ± 0.216; MSU + P140 = 4.28 ± 0.535); *p* = 0.0013 when compared MSU vs. MSU + P140).

These findings suggest that P140 post-treatment affects neutrophil survival in the joints and could affect the neutrophil cell death program critical for the optimal and efficient resolution of MSU-induced inflammation.

### 3.4. Treatment with P140 Affects the Survival of Neutrophils In Vivo and Promotes the Production of a Pro-Resolving Mediator In Vitro

Accumulating data indicate that neutrophils are the most abundant cells among the leukocytes, which play a crucial role in the acute inflammatory response induced by MSU crystals [5]. After neutrophil mobilization from blood to the tissue releases several molecules with cytotoxic potential that contribute significantly to an extensive and inappropriate tissue injury [32]. The injection of MSU crystals triggers an inflammatory response with a significant accumulation of neutrophils in the joints with a peak between 12–15 h post-induction [4]. Using the protocol described above (P140 given 6 h after MSU injection with a sacrifice occurring 9 h later, thus maintaining the delay of 15 h after MSU induction), we observed increase apoptosis of neutrophils, which was not found when ScP140 was used as control peptide (Figure 4A). This effect of P140 on neutrophil survival was confirmed using isolated neutrophils from healthy human donors. Neutrophils stimulated with P140 increased caspase-3 activity (Figure 4B), associated with NF-κB inhibition on MSU-challenged cells (Figure 4C). Both effects observed on neutrophils are critical events for induction of neutrophil’s apoptosis, as already described previously [33]. Of note, apoptosis of neutrophils is an essential and safe event during the resolution of inflammation [34]. We further tested the effect of P140 on the expression of annexin A1, a glucocorticoid-regulated protein known to promote inflammation resolution by inducing neutrophil apoptosis [35]. When comparing MSU-stimulated neutrophils treated or not with P140, a clear trend towards a higher expression of annexin A1 was observed upon P140 treatment (Figure 4D). These results reinforce the hypothesis that in our experimental model of gout, the beneficial effect of P140 could, at least in part, results from its ability to decrease the lifespan of activated neutrophils, which are recognized contributors to synovitis, local joint destruction, and systemic inflammation in gout.

### 3.5. Effect of P140 on the Protein Levels of Autophagy Markers in Neutrophils Isolated from Human Blood and Stimulated In Vitro with MSU

Consistent evidence indicates that P140 exerts its beneficial activity by modulating the abnormal autophagy process [19,20,36]. Autophagy is a tightly regulated mechanism that allows cells to self-renew by breaking down misfolded, overproduced, or damaged proteins. This process plays a crucial role in cellular homeostasis and immune responses [37,38]. In neutrophils, autophagy is essential for significant functions, including degranulation, reactive oxygen species production, and release of neutrophil extracellular traps (NETs) under stress and infection [39,40]. Previous studies have shown that P140 could correct dysfunctional chaperone-mediated autophagy (CMA) and macroautophagy in different in vitro and in vivo contexts [11,12,13,21,41], especially in vitro, in primary spleen cells of MRL/lpr lupus mice [19,20]. This prompted us to investigate the effect of P140 on autophagy in MSU-activated neutrophils (Figure 5; Appendix A). These experiments led us to observe that in vitro human neutrophils treated with MSU seem to express higher levels of HSPA8 and covalent ATG5-ATG12 conjugate compared to neutrophils non-activated by MSU. Upon treatment with P140, the increased levels of expression of HSPA8 and ATG5-ATG12 conjugate return to basal levels (Figure 5A,B). 

Together, these findings indicate that MSU stimulus activates CMA and macroautophagy processes in neutrophils and suggest that treatment with P140 could downregulate hyperactive autophagy processes in neutrophils during gout inflammation.

## 4. Discussion

In this study, we characterized the effects of the P140 peptide in the resolution of acute inflammation induced by MSU crystals during gout development in mice. The therapeutic benefit of P140 has been demonstrated in two induced mouse models of acute gout. We observed that treatment with the immunomodulatory peptide P140 promoted inflammation reduction by reducing neutrophils in the inflamed tissue, consequently decreasing the pro-inflammatory mediators released, and increasing apoptosis of neutrophils. The beneficial effect of P140 was observed even 10 h post-MSU crystal induction. Our findings provide a new therapeutic candidate that is not immunosuppressive [42] and might be promising for gout patients.

During the development of gout, resident macrophages are the primary sources of IL-1β secretion, which play a critical role in the recruitment and massive infiltration of neutrophils [22]. Subsequently, neutrophils predominate at the inflammatory site and represent a significant cell population contributor to the pathology of gout through the release of pro-inflammatory mediators, such as IL-1β, TNF-α, or NETs and proteases [43]. We found here that P140 treatment decreased some of the pro-inflammatory cytokines in the periarticular tissue in both in vivo experimental models of gout, indicating that P140 can attenuate gout inflammation via modulation of neutrophil biology function. Although we have observed an increased number of neutrophils in the blood and MPO release in spleen after MSU injection and P140 treatment, we demonstrated a reduction in MPO release during neutrophil degranulation in periarticular tissue in vivo upon treatment with P140. There is a hypothesis that P140 accumulates in the spleen [36], which could affect circulating neutrophils and regulate their recruitment to the site of inflammation. This strongly suggests that neutrophil activity might be directly or indirectly affected by P140. Studies have shown that activation of neutrophils by MSU crystals leads to apoptosis inhibition and promotes NET formation of NETs [43,44,45]. The formation of NETs is accompanied by the release of enzymes, such as MPO, that encourage tissue damage [11]. Although we did not explore the effect of P140 on NET formation in gout, a previous study showed that NETosis, a specific kind of cell death that ensues when neutrophils extrude NETs, was decreased in human neutrophils during in vitro incubation in the presence of P140 [18]. Thus, the attenuation of NETosis could be favoured by the induction of the apoptotic process induced by P140. Apoptosis of neutrophils is the main event that contributes to the resolution of inflammation to restore tissue homeostasis. It has been highly suggested that a pharmacological strategy that could be used to enhance the resolution of inflammation by redirecting neutrophils to apoptosis might be the novel and potential solution to treating many inflammatory diseases [46]. It has also been shown that decreasing the recruitment of neutrophils to the inflammatory site reduces inflammation and promotes the resolution of the gout attack [33]. The reduction of neutrophil migration into the site of inflammation could explain the improvement in gout symptoms in our models, a feature that we also found in models of asthma [21] and lupus [18]. Here, we confirm that P140 reduces the migration of human neutrophils in vitro when stimulated by LPS and MSU crystals. 

On the other hand, the induction of neutrophil’s apoptosis makes it possible to limit the degranulation of neutrophils. Neutrophil apoptosis promotes extracellular matrix clearance through the uptake of apoptotic compartments by macrophages and ultimately contributes to the resolution of inflammation [47]. Although we have not seen significant result, a trend towards the reduction of neutrophils was observed in the knee cavity along with increased number of neutrophil apoptosis. This may suggest that P140 might reduce the time that inflammation persist; however, we have not evaluated resolution effect as a function of time of P140. Our results highlight a potential pro-apoptotic effect of P140 on isolated human neutrophils. Our experiments revealed an increase in caspase-3 activity in isolated human neutrophils treated with P140. In addition, the decrease in the expression of NF-κB in human neutrophils isolated after treatment with P140 is complementary to our observations at the level of caspase-3, since this protein allows the activation of the transcription of anti-apoptotic genes such as Bcl-2, Traf1, and Traf2 [48]. Corroborating our results, dysfunctional autophagy has been associated with an increased apoptosis suggesting that modulation of autophagy might be the mechanism by which P140 increased caspase-3 and apoptosis [49].

The pro-resolving effect of the therapeutic peptide P140 has never been examined as extensively. In this study, we show data suggesting increased expression of annexin A1 in human neutrophils stimulated with MSU crystals and treated with P140. Annexin A1 plays a crucial role on resolution of inflammation in a model of gout by increasing neutrophil apoptosis [50]. Annexin A1 in its intact form (37 kDa) is a protein regulated by the glucocorticoid signalling pathway and has anti-inflammatory and pro-resolving properties [35,50]. In neutrophils, annexin A1 is cleaved by proteinase 3 and elastase, thereby generating the inactive form of annexin A1 (33 kDa) [51]. Active annexin A1 (37 kDa) can be externalized to the neutrophil membrane surface and bind formyl peptide 2 receptors. Studies have shown that annexin A1 stimulates neutrophil apoptosis and also inhibition of NF-κB, which corroborates our previous results [51,52]. Interestingly, annexin A1 can activate the phosphoinositide 3-kinase signalling pathway inducing inhibition of BECLIN-1, which ultimately leads to inhibition of the macroautophagy process [53]. It is also important to mention that annexin A family regulates the formation of vesicular lipid membrane and cell exocytosis that are also mechanisms associated with autophagy [53]. Moreover, BECLIN-1 [54] and ATG5 [55] have been shown to be cleaved by caspase-3, contributing to the shift of cellular response towards apoptosis. Our results in isolated human neutrophils show an inhibitory effect of P140 on both heat shock protein HSPA8 and ATG5-ATG12 levels of expression, signalling CMA and macroautophagy. At this stage, however, further in-depth investigation is required to establish the link between the synthesis of annexin A1 and the inhibition of autophagy during treatment with P140.

In our studies, we have not observed a significant modulation of the inflammasome in macrophages, which drives IL-1β release and neutrophil activation and recruitment. It has been reported that the activity of neutrophil-derived proteases plays a more dominant role in the acute inflammation induced by IL-1β in the gout context, suggesting that caspase-1 plays a minor role. Although we do not exclude that the effect of P140 on neutrophils would be involved, it has been suggested that modulation of inflammasome pathways is not an effective therapeutic strategy in pathologies in which there is massive neutrophil infiltration [56].

Taken together, our study provides new data on the effect of the P140 peptide on neutrophils. The results presented here highlight a potential effect of P140 on neutrophil recruitment and cell death by apoptosis in MSU crystals induced acute inflammation in gout. Many studies point to these mechanisms of action as beneficial in promoting the resolution of inflammation in gout. For the first time, we demonstrated the potential modulation effect of P140 peptide in gouty inflammation that may represent a potent pro-resolving mediator. This therapeutic peptide, which is currently evaluated in phase III clinical trials for lupus, could be advantageously repositioned. The routes that this peptide was administered in lupus patients and normal individuals during trial were s.c. and i.v., and we could hypothesize the same routes for patients with gout. Moreover, although the effect of P140 in controlling gouty inflammation is a promising approach, no evidence of any effect on uric acid levels in serum was observed and is not expected. The effects of P140 have been investigated in other inflammatory and autoimmune disorders [12,13,21,41] and added to the toolbox that fulfils the next-generation pipeline of drugs that may be used to treat patients with gout. 

## Figures and Tables

**Figure 1 cells-11-03709-f001:**
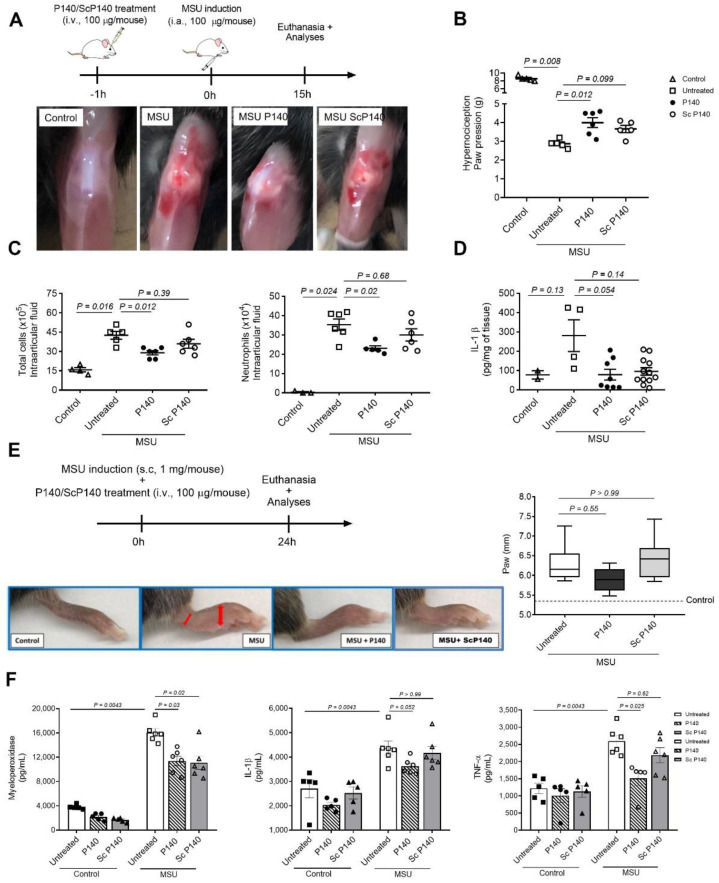
**In vivo effect of P140 peptide on two mouse models of gout**. (**A**) First mouse model: schedule of treatment (upper panel) and representative images of inflammation site (knee) of control (no MSU—treated), MSU challenged, pre-treated P140 + MSU, and pre-treated Scrambled P140 (ScP140) + MSU mice. In gout (MSU) P140-treated mice the reduction of oedema and inflammation at the level of the knee (lower panel) is depicted. (**B**) Mechanical hypernociception (pain) assessed 15 h after injection of MSU crystals (100µg intraarticular–i.a.) by electronic pressure-meter measuring paw withdrawal by flexion of the femorotibial joint in mice treated, or not, with P140 and ScP140 control. (**C**) Number of total cells and neutrophils count in the articular cavity. (**D**) Levels of IL-1β measured by ELISA 15 h after MSU crystal injection in periarticular tissues. (**E**) Second mouse model: schedule of treatment (upper panel) representative images (lower panel) and size (right panel) of hind paws of control (no MSU-treated), MSU challenged, co-treated P140 + MSU, and co-treated ScP140 + MSU mice (lower panel). Size of hind paws was measured by calliper and expressed in millimetres (mm). (**F**) Levels of MPO and pro-inflammatory cytokines IL-1β and TNF-α, measured 24 h after injection of MSU crystals in paw tissues. These results are representative of 2 independent experiments. The ScP140 analogue was used as negative control in the same conditions as those used with P140. Data are mean ± SEM of 5–6 mice per group, *p* values were calculated using Mann–Whitney or Kruskal–Wallis tests following Dunn’s post hoc test.

**Figure 2 cells-11-03709-f002:**
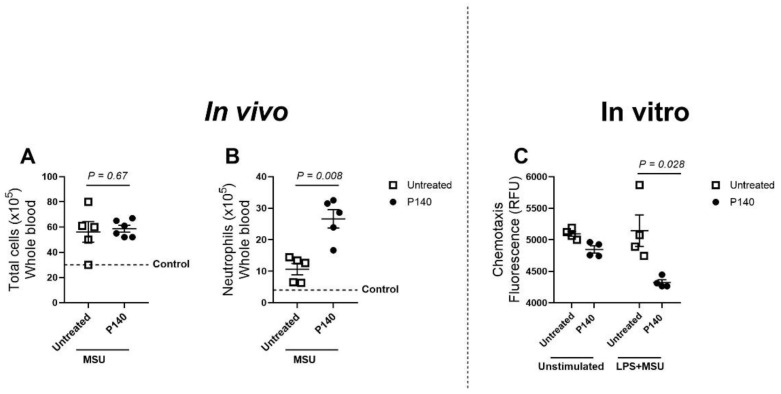
**In vivo and in vitro effect of P140 on the presence of neutrophils in the peripheral blood in a model of gout.** (**A**) Total cell counts measured by haemocytometer and (**B**) number of neutrophils measured by smear count determined in the peripheral blood of mice 15 h after MSU injection. Mice were pre-treated with P140 1 h before MSU crystal injection. (**C**) In vitro effect of P140 peptide on the neutrophil migration isolated from human peripheral blood and measured by Boyden chamber. CXCL8 was used to induce cell migration. Neutrophils were stimulated or not with LPS and MSU. The results are expressed in relative fluorescence units (RFU). Data are reported as mean ± SEM of 5–6 mice per group. *p* value was calculated using Mann-Whitney test. n.s = non-statistical.

**Figure 3 cells-11-03709-f003:**
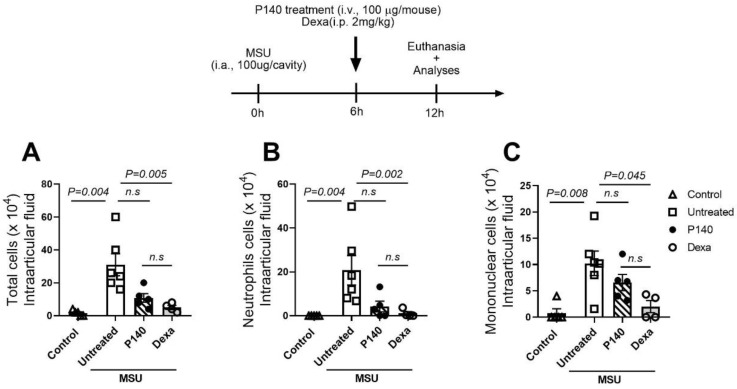
**Effect of P140 peptide and dexamethasone on the number of neutrophils and mononuclear cells in the intra-articular cavity of the knee of gouty mice**. Mice were post-treated with P140 peptide (100 µg/mouse; i.v.) or dexamethasone (2 mg/kg i.p.) 6 h after MSU injection. Cells were harvested from the i.a. cavity 15 h post-MSU induction. (**A**) The number of total cells, (**B**) neutrophils, and (**C**) mononuclear leukocytes were determined by Neubauer chamber and cytospin counting. Data are mean ± SEM of 5–6 mice per group, *p* values were calculated using Mann–Whitney or Kruskal–Wallis tests following Dunn’s post hoc test. n.s = non-statistical.

**Figure 4 cells-11-03709-f004:**
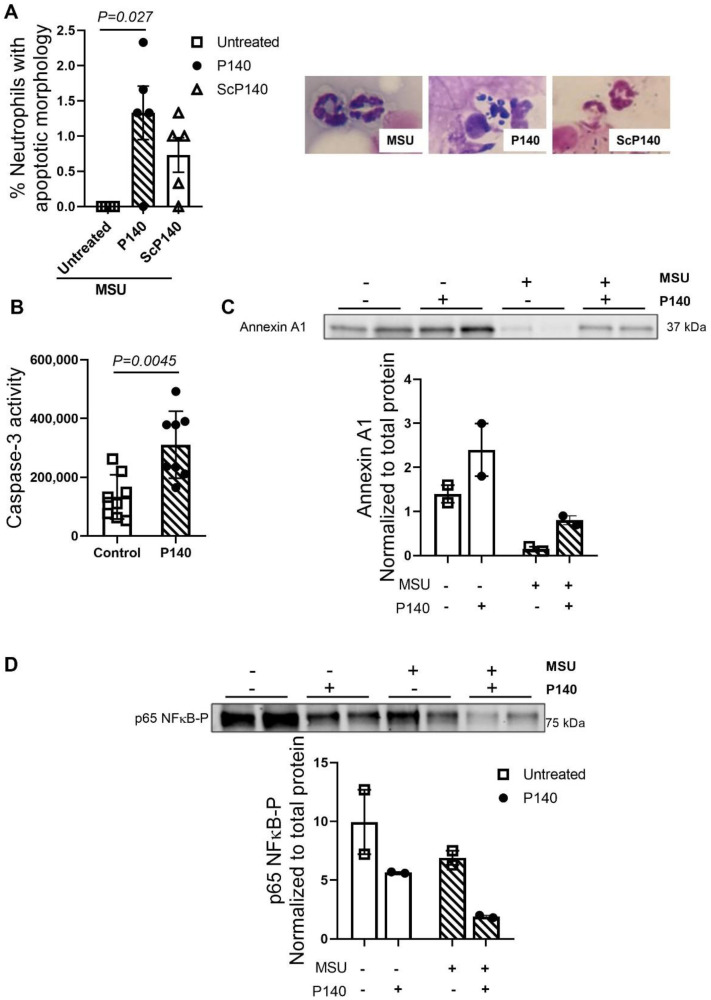
**Effect of P140 peptide on the survival of neutrophils isolated from human peripheral blood.** (**A**) In vivo percentage of neutrophils with apoptotic morphology measured in untreated, P140-, and ScP140-treated mice after gout induction. Pictures were representative of neutrophils collected from synovial fluid, stained, and count for distinctive apoptotic morphology. (**B**) Caspase-3 activity was measured in human neutrophils 4 h after P140 incubation. (**C**) Representative western blot images and protein quantification normalized to total protein for phosphorylated p65 NFκB protein measured in neutrophils upon MSU stimulation and P140 treatment. (**D**) Representative western blot images and protein quantification normalized to total protein for annexin A1 in neutrophils upon MSU stimulation and P140 treatment. Data are mean ± SEM. *p* values were calculated using Mann–Whitney or Kruskal–Wallis tests following Dunn’s post hoc test.—means absence and + means presence.

**Figure 5 cells-11-03709-f005:**
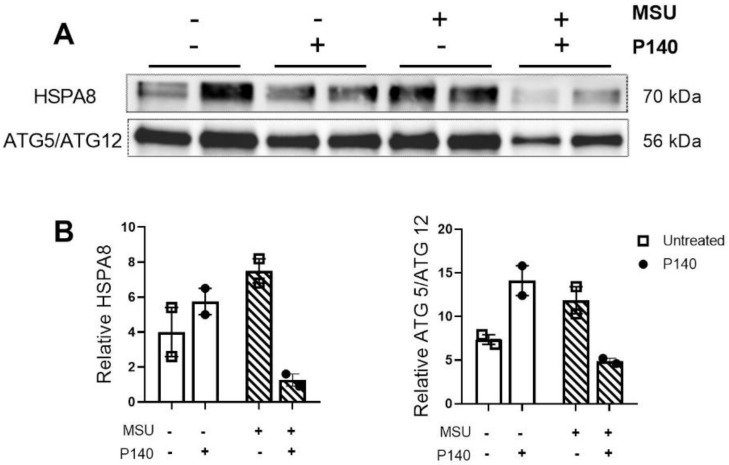
**Effect of P140 peptide on the expression of ATG5-ATG12 complex and HSPA8 in neutrophils isolated from human peripheral blood**. (**A**) Representative western blot images and (**B**) protein quantification normalized to total protein for ATG5-ATG12 and HSPA8 upon stimulation of neutrophils by MSU crystals and P140 treatment. Data are mean ± SEM.—means absence and + means presence.

## Data Availability

All data needed to evaluate the conclusions of the paper are present in the paper. Additional data related to this paper may be requested from the corresponding author.

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
