# Peer review of "The Therapeutic Effect of Phosphopeptide P140 Attenuates Inflammation Induced by Uric Acid Crystals in Gout Arthritis Mouse Model"

_cells, 2022, doi:10.3390/cells11233709_

Round 1
Reviewer 1 Report
This paper reports that phosphopeptide P140 reduces monosodium urate crystal-induced inflammation in experimental models. In addition, the authors suggest that P140 administration increases neutrophil apoptosis in vivo and promotes the production of pro-resolving mediator in vitro.
The study is interesting, however some issues have to be addressed by the authors in order to improve the manuscript.
Comments and suggestions are list below:
1. In “Materials and methods” section the experimental design is not clear. The manuscript has to be supplemented with the following information: number of animals, how animals were divided into groups, if animals were randomized. In addition the authors should better explain when they treat animals with P140 and specify when they use a prophylactic or a therapeutic protocol. How was the sample size calculated?
2. The results report the analysis of the spleen, but this is not described in the “materials and methods” section.
3 3. Page 7, please rephrase lines 279-282.
4. Since generally the studies on intraarticularly injected MSU crystals consider the effect at 48 h where there is maximum inflammation, the authors should explain in the text why they decided to consider the effect at 15 h? The same, for the therapeutic treatment after 6 hours from the injection of the crystals.
5. Paragraph 3.1, the administration of P140 never significantly reduces the production of IL-1beta, this has to be corrected and specified in the results and in the discussion. Authors should evaluate adhesion molecule expression and local chemokine production (including IL-8) that play a key role in the neutrophil recruitment.
6. The histological analysis of the paw of the mice simultaneously injected with MSU and P140 has to be performed and accompanied by the histopathology scores.
7. How do the authors explain the increase in the number of neutrophils in the blood and spleen in animals injected with MSU and treated with P140?
8. Paragraph 3.3 is titled “P140 doesn’t affect neutrophil accumulation but instead increases neutrophils’ apoptosis when an inflammatory status occurs”, but apoptosis has not yet been evaluated here.
9. The authors do not demonstrate that P140 administration reduced inflammation in treated mice with P140 after the establishment of inflammation induced by MSU crystal injection in the knee (6 h after MSU injection). At least the graph and the significance of the nociception should be shown.
10. In figure 4, were NF-κB inhibition and increase of annexin A1 levels induced by p140 treatment of neutrophils significant?
11. In figure 5, the significances should be added. How do the authors explain the increase of HSPA8 and ATG5-ATG12 induced by P140 in the absence of MSU crystals and then the decrease in the presence of crystals?
12. Dothe authors hypothesize a route of administration of this molecule in patients with gout? Write in the “discussion” section.
Author Response
Point-by-point response to Reviewers comments
We thank the Editor and Reviewers for critically examining our article and providing invaluable suggestions. We have responded to each comment and clarified/amended every part with the appropriate information. The statements addressing the Reviewers’ comments point-by-point are listed below and alterations in the main text are in yellow highlight in revised version. We highly appreciate your consideration and further process.
Review 1
This paper reports that phosphopeptide P140 reduces monosodium urate crystal-induced inflammation in experimental models. In addition, the authors suggest that P140 administration increases neutrophil apoptosis in vivo and promotes the production of pro-resolving mediator in vitro.
The study is interesting, however some issues have to be addressed by the authors in order to improve the manuscript.
Comments and suggestions are list below:
- In “Materials and methods” section the experimental design is not clear. The manuscript has to be supplemented with the following information: number of animals, how animals were divided into groups, if animals were randomized. In addition the authors should better explain when they treat animals with P140 and specify when they use a prophylactic or a therapeutic protocol. How was the sample size calculated?
We thank the reviewer for the comment. We have included a new section on materials and methods called experimental design with all information required by the reviewer on page 3.
- The results report the analysis of the spleen, but this is not described in the “materials and methods” section.
We thank the reviewer for the comment. We have included in the materials and methods how spleen was processed in the revised manuscript on page 4.
- Page 7, please rephrase lines 279-282.
We thank the reviewer for the comment. We have rephrased as suggested.
- Since generally the studies on intraarticularly injected MSU crystals consider the effect at 48 h where there is maximum inflammation, the authors should explain in the text why they decided to consider the effect at 15 h? The same, for the therapeutic treatment after 6 hours from the injection of the crystals.
In our hands, as previously reported (Amaral FA, et al. Arthritis Rheum. 2012 Feb;64(2):474-84. doi: 10.1002/art.33355); (Vieira AT, et al J Leuko Biol. 2016 J Leukoc Biol. 2017 Jan;101(1):275-284. doi: 10.1189/jlb.3A1015-453RRR.); (Galvão I, et al Eur J Immunol. 2017 Mar;47(3):585-596. doi: 10.1002/eji.201646551) intra-articular injection of MSU crystals induced a time-dependent influx of leukocytes into the knee joint of mice, which was significant at 6 h and peaked between 12 h and 15h. Infiltrating leukocytes were mainly neutrophils. The number of neutrophils decreased drastically at 24 h, and they were almost absent at 48 h after injection of MSU crystals. The number of mononuclear cells also increased at 12 h and decreased thereafter. At 48 h, mononuclear cells were the main leukocytes found in the joint. We have chosen 15h to evaluate the treatment effect since this is the time where we observed the maximum neutrophil infiltration and the therapeutic treatment at 6h since there are a significant influx of neutrophils to the joint at this time. We have included this information in the result section as suggested by the reviewer on page 7.
- Paragraph 3.1, the administration of P140 never significantly reduces the production of IL-1beta, this has to be corrected and specified in the results and in the discussion. Authors should evaluate adhesion molecule expression and local chemokine production (including IL-8) that play a key role in the neutrophil recruitment.
We thank the reviewer for this comment and apologized for the confused information. We do see significant reduction of IL-1B after administration of P140. In figure 1 we have observed that P140 reduce IL-1B levels in the knee model (Figure 1D) and paw model (Figure 1F) when compared with untreated group. We have not seen significant reduction of this cytokines on the group treated with the scrambled peptide (SCP140) used as a peptide control. We have clarified this message in the text of the revised manuscript. In addition, we have evaluated the local production of CXCL1(KC) in the periarticular tissue on gout model, and this result is shown in the text line 299-300.
- The histological analysis of the paw of the mice simultaneously injected with MSU and P140 has to be performed and accompanied by the histopathology scores.
We agree that histological analysis of the paw and the histopathology score would be more informative. However, to do this, we would need to perform additional experiments only for histopathology analysis, which is not possible at this stage. Anyhow, clinical data displayed on figure 1 suggested that P140 treatment ameliorates MSU induced inflammation by improvement of oedema and reduction of MPO. Additionally, we have provided the representative histology images of the knee in the supplementary material, where we observed that treatment with P140 reduced the amount of inflammatory infiltrate (black arrows) compared with untreated group.
- How do the authors explain the increase in the number of neutrophils in the blood and spleen in animals injected with MSU and treated with P140?
The capacity of the bone marrow to quickly increase the number of neutrophils output in response to inflammatory signal is an explanation of why the numbers of neutrophils in the blood are increased after MSU crystals injection. Moreover, there is a hypothesis that P140 might accumulate in the spleen (Page N et al, Ann Rheum Dis. 2011 May;70(5):837-43. doi: 10.1136/ard.2010.139832), which could affect circulating neutrophils and regulate their recruitment to the site of inflammation. We have not investigated this in the present model. We have included this information on the revised manuscript on the discussion section page 14.
- Paragraph 3.3 is titled “P140 doesn’t affect neutrophil accumulation but instead increases neutrophils’ apoptosis when an inflammatory status occurs”, but apoptosis has not yet been evaluated here.
We have changed the title as suggested.
- The authors do not demonstrate that P140 administration reduced inflammation in treated mice with P140 after the establishment of inflammation induced by MSU crystal injection in the knee (6 h after MSU injection). At least the graph and the significance of the nociception should be shown.
We thank the reviewer for the comment. Although not significant, we observed a decreased number of neutrophil accumulations in the knee cavity. Moreover, there was a reduction of hypernociception. The results were included in the text as mean ± SEM line 386-387.
- In figure 4, were NF-κB inhibition and increase of annexin A1 levels induced by p140 treatment of neutrophils significant?
To perform the western blot, samples had to be pooled in order to load enough protein to evaluate NFKB and Annexin A1 expression. Because of this technical limitation, as observed in figure 4, we only have n equals 2 per group, which makes the statistical analysis not possible.
- In figure 5, the significances should be added. How do the authors explain the increase of HSPA8 and ATG5-ATG12 induced by P140 in the absence of MSU crystals and then the decrease in the presence of crystals?
As above (comment 10) samples had to be pooled in order to load enough protein to evaluate HSPA8 and ATG5-ATG12 expression. As a consequence, we only have n equals 2 per group, which makes the statistical analysis not possible.
Reviewer 2 Report
I enjoyed reading this manuscript and highly commend the authors for their careful experimental approach. As a clinician working with gout and managing patients, I am always excited about the perspective of new therapeutic approaches.
I have the following comments:
-Introduction, line 61: It is not accurate to say that “The current therapies are ineffective in all patients”. We do have quite effective therapies. However, these are not always tolerated or comorbidities/contraindications might [preclude them from being used in a large proportion of those with gout.
-I was surprised that the authors presented results and conclusions of the study in the last paragraph of the introduction
-I would emphasize in the discussion that this is a promising approach for the management of gout flares, without an expected effect on serum urate levels or control of hyperuricemia.
-Could p140 peptide also be investigated in other autoinflammatory or autoimmune disorders where neutrophils play a prominent role such as ANCA-associated vasculitis, Still’s disease? I know I am inviting the authors to speculate too much, but I wonder if it could be considered in future directions.
Author Response
Point-by-point response to Reviewers comments
We thank the Editor and Reviewers for critically examining our article and providing invaluable suggestions. We have responded to each comment and clarified/amended every part with the appropriate information. The statements addressing the Reviewers’ comments point-by-point are listed below and alterations in the main text are in yellow highlight in revised version. We highly appreciate your consideration and further process.
Review 2
I enjoyed reading this manuscript and highly commend the authors for their careful experimental approach. As a clinician working with gout and managing patients, I am always excited about the perspective of new therapeutic approaches.
I have the following comments:
-Introduction, line 61: It is not accurate to say that “The current therapies are ineffective in all patients”. We do have quite effective therapies. However, these are not always tolerated or comorbidities/contraindications might [preclude them from being used in a large proportion of those with gout.
We agree with this remark and have changed the introduction as correctly requested by the reviewer.
-I was surprised that the authors presented results and conclusions of the study in the last paragraph of the introduction.
We agree that this style can be surprising and takes away the suspense of the section of data that follows. It is classically requested in the Instructions to Authors, however, we have changed accordingly.
-I would emphasize in the discussion that this is a promising approach for the management of gout flares, without an expected effect on serum urate levels or control of hyperuricemia.
We thank the reviewer for the comment. We have emphasized in the discussion what was suggested.
-Could p140 peptide also be investigated in other autoinflammatory or autoimmune disorders where neutrophils play a prominent role such as ANCA-associated vasculitis, Still’s disease? I know I am inviting the authors to speculate too much, but I wonder if it could be considered in future directions.
Yes, indeed, the effect of P140 in other inflammatory and autoimmune disorders has been investigated in our laboratory. The results have been published:
-Brun, S., Schall, N., Bonam, S.R, Bigaut, K., Mensah-Nyagan, A.G., de Sèze, J. & Muller, S. (2018) An autophagy-targeting peptide to treat chronic inflammatory demyelinating polyneuropathies. J. Autoimmunity 92:114-125.
-Li, B., Wang, F., Schall, N. & Muller, S. (2018) Rescue of autophagy and lysosome defects in salivary glands of MRL/lpr mice by a therapeutic phosphopeptide. J. Autoimmunity 90:132-145.
-Voynova, E., Lefebvre, F., Qadri, A. & Muller, S. (2020) Correction of autophagy impairment inhibits pathology in the NOD.H-2h4 mouse model of primary Sjögren’s syndrome. J. Autoimmunity 108:102418
-Daubeuf, F., Schall, N., Petit-Demoulière, N., Frossard, N. & Muller, S. (2021) An autophagy modulator peptide prevents lung function decrease and corrects established inflammation in murine models of airway allergy. Cells 10:2468
-Retnakumar, S.V., Geesala, R., Bretin, A., Tourneur-Marsille, J., Ogier-Denis, E., Maretzky, T., Nguyen, H.T.T & Muller, S. (2022) Targeting the endo-lysosomal autophagy pathway to treat inflammatory bowel diseases. J. Autoimmunity 128:102814
-Akiyama, K., Aung, K.T., Talamini, L., Huck, O., Kuboki, T. & Muller, S. (2022) Therapeutic effects of peptide P140 in a mouse periodontitis model. Cell. Mol. Life Sci, 79:518
Note: Neutrophils are important in asthma and effectively we showed some effect on this subpopulation (Daubeuf et al., 2021). The effect of P140 was never tested in the context of ANCA-associated vasculitis or Still’s disease. P140 exerts some effects on NETosis (Bendorius, M., Neeli, I., Wang, F., Bonam, S.R., Dombi, E., Buron, N., Borgne-Sanchez, A., Poulton, J., Radic, M. & Muller, S. (2018) The mitochondrion-lysosome axis in adaptive and innate immunity: Effect of lupus regulator peptide P140 on mitochondria autophagy and NETosis. Front. Immunol. 9:2158).
Round 2
Reviewer 1 Report
The authors have modified the manuscript as suggested, however some other minor changes should be done.
1. Page 3, line 125 the verb is missing
2. Page 7, lines 282-285 the sentence needs to be rearranged
Author Response
We thank the reviewer for critically examine our manuscript. Please see bellow our answers:
- Page 3, line 125 the verb is missing
We have added the missing verb as required as seen bellow and in the revised manuscript.
"In the second set of experiment a single injection of MSU crystals (s.c. 3mg) into the hind paw was done. "
- Page 7, lines 282-285 the sentence needs to be rearranged
The sentence has been changed as required as seen bellow and in the revised manuscript.
"As previously described, the injection of MSU crystals in the knee joint of mice induces a significant leukocyte recruitment at 6 hours predominantly consistent by neutrophil that reach maximum number at 15 hours after MSU injection. Moreover, neutrophil-dependent tissue injury and articular dysfunction is observed during inflammatory response induced by MSU crystals "